# CENTRAL SERVER FREE FEDERATED LEARNING OVER SINGLE-SIDED TRUST SOCIAL NETWORKS

## ABSTRACT

Federated learning has become increasingly important for modern machine learning, especially for data privacy-sensitive scenarios. Existing federated learning mostly adopts the central server-based architecture or centralized architecture. However, in many social network scenarios, centralized federated learning is not applicable (e.g., a central agent or server connecting all users may not exist, or the communication cost to the central server is not affordable). In this paper, we consider a generic setting: 1) the central server may not exist, and 2) the social network is unidirectional or of single-sided trust (i.e., user A trusts user B but user B may not trust user A). We propose a central server free federated learning algorithm, named Online Push-Sum (OPS) method, to handle this challenging but generic scenario. A rigorous regret analysis is also provided, which shows interesting results on how users can benefit from communication with trusted users in the federated learning scenario. This work builds upon the fundamental algorithm framework and theoretical guarantees for federated learning in the generic social network scenario.

## 1 INTRODUCTION

Federated learning has been well recognized as a framework able to protect data privacy Konečný et al. (2016); Smith et al. (2017a); Yang et al. (2019). State-of-the-art federated learning adopts the centralized network architecture where a centralized node collects the gradients sent from child agents to update the global model. Despite its simplicity, the centralized method suffers from communication and computational bottlenecks in the central node, especially for federated learning, where a large number of clients are usually involved. Moreover, to prevent reverse engineering of the user's identity, a certain amount of noise must be added to the gradient to protect user privacy, which partially sacrifices the efficiency and the accuracy Shokri and Shmatikov (2015).

To further protect the data privacy and avoid the communication bottleneck, the decentralized architecture has been recently proposed Vanhaesebrouck et al. (2017); Bellet et al. (2018), where the centralized node has been removed, and each node only communicates with its neighbors (with mutual trust) by exchanging their local models. Exchanging local models is usually favored to the data privacy protection over sending private gradients because the local model is the aggregation or mixture of quite a large amount of data while the local gradient directly reflects only one or a batch of private data samples. Although advantages of decentralized architecture have been well recognized over the state-of-the-art method (its centralized counterpart), it usually can only be run on the network with *mutual trusts*. That is, two nodes (or users) can exchange their local models only if they trust each other reciprocally (e.g., node A may trust node B, but if node B does not trust node A, they cannot communicate). Given a social network, one can only use the edges with mutual trust to run decentralized federated learning algorithms. Two immediate drawbacks will be: (1) If all mutual trust edges do not form a connected network, the federated learning does not apply; (2) Removing all single-sided edges from the communication network could significantly reduce the efficiency of communication. These drawbacks lead to the question: *How do we effectively utilize the single-sided trust edges under decentralized federated learning framework?*

In this paper, we consider the social network scenario, where the centralized network is unavailable (e.g., there does not exist a central node that can build up the connection with all users, or the centralized communication cost is not affordable). We make a minimal assumption on the social

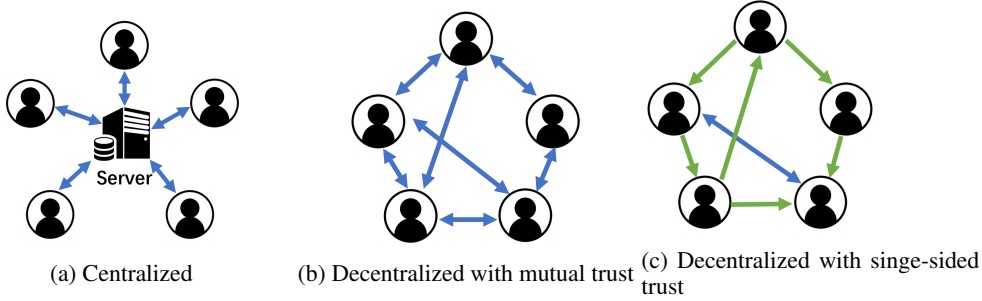

(a) Centralized      (b) Decentralized with mutual trust    (c) Decentralized with singe-sided trust

Figure 1: Different types of architectures.

network: The data may come in a streaming fashion on each user node as the federated learning algorithm runs; the trust between users may be single-sided, where user A trusts user B, but user B may not trust user A ("trust" means "would like to send information to").

For the setting mentioned above, we develop a decentralized learning algorithm called online push-sum (OPS) which possesses the following features:

- Only models rather than local gradients are exchanged among clients in our algorithm. This scheme can reduce the risk of exposing clients' data privacy Aono et al. (2017).

- Our algorithm removes some constraints imposed by typical decentralized methods, which makes it more flexible in allowing arbitrary network topology. Each node only needs to know its out neighbors instead of the global topology.

- We provide the rigorous regret analysis for the proposed algorithm and specifically distinguish two components in the online loss function: the adversary component and the stochastic component, which can model clients' private data and internal connections between clients, respectively.

**Notation** We adopt the following notation in this paper:

- For random variable $\boldsymbol{\xi}_t^{(i)}$ subject to distribution $D_t^{(i)}$, we use $\Xi_{n,T}$ and $\mathcal{D}_{n,T}$ to denote the set of random variables and distributions, respectively:

$$\Xi_{n,T} = \left\{\boldsymbol{\xi}_t^{(i)}\right\}_{1 \leq i \leq n, 1 \leq t \leq T}, \quad \mathcal{D}_{n,T} = \left\{D_t^{(i)}\right\}_{1 \leq i \leq n, 1 \leq t \leq T}.$$

Notation $\Xi_{n,T} \sim \mathcal{D}_{n,T}$ implies $\boldsymbol{\xi}_t^{(i)} \sim D_t^{(i)}$ for any $i \in [n]$ and $t \in [T]$.

- For a decentralized network with $n$ nodes, we use $\mathbf{W} \in \mathbb{R}^{n \times n}$ to present the confusion matrix, where $W_{ij} \geq 0$ is the weight that node $i$ sends to node $j$ $(i, j \in [n])$. $\mathcal{N}_i^{\text{out}} = \{j \in [n] : W_{ij} > 0\}$ and $\mathcal{N}_i^{\text{in}} = \{k \in [n] : W_{ki} > 0\}$ are also used for denoting the sets of in neighbors of and out neighbors of node $i$ respectively.

- Norm $\|\cdot\|$ denotes the $\ell_2$ norm $\|\cdot\|_2$ by default.

## 2 RELATED WORK

The concept of *federated learning* was first proposed in McMahan et al. (2016), which advocates a novel learning setting that learns a shared model by aggregating locally-computed gradient updates without centralizing distributed data on devices. Early examples of research into federated learning also include Konečný et al. (2015; 2016), and a widespread blog article posted by Google AI McMahan and Ramage (2017). To address both statistical and system challenges, Smith et al. (2017b) and Caldas et al. (2018) propose a multi-task learning framework for federated learning and its related optimization algorithm, which extends early works SDCA Shalev-Shwartz and Zhang (2013); Yang (2013); Yang et al. (2013) and COCOA Jaggi et al. (2014); Ma et al. (2015); Smith et al. (2016) to the federated learning setting. Among these optimization methods, *Federated Averaging* (FedAvg), proposed by McMahan et al. (2016), beats conventional synchronized mini-batch SGD regarding communication rounds as well as converges on non-IID and unbalanced data. Recent rigorous theoretical analysis Stich (2018); Wang and Joshi (2018); Yu et al. (2018); Lin et al. (2018) shows that FedAvg is a special case of averaging periodic SGD (also called "local SGD") which allows nodes

to perform local updates and infrequent synchronization between them to communicate less while converging quickly. However, they cannot be applied to the single-sided trust network (asymmetric topology matrix).

Decentralized learning is a typical parallel strategy where each worker is only required to communicate with its neighbors, which means the communication bottleneck (in the parameter server) is removed. It has already been proved that decentralized learning can outperform the traditional centralized learning when the worker number is comparably large under a poor network condition Lian et al. (2017). There are two main types of decentralized learning algorithms: fixed network topology He et al. (2018), and time-varying Nedić and Olshevsky (2015); Lian et al. (2018) during training. Wu et al. (2017); Shen et al. (2018) shows that the decentralized SGD would converge with a comparable convergence rate to the centralized algorithm with less communication to make large-scale model training feasible. Li et al. (2018) provides a systematic analysis of the decentralized learning pipeline.

Online learning has been studied for decades. It is well known that the lower bounds of online optimization methods are $\mathcal{O}(\sqrt{T})$ and $\mathcal{O}(\log T)$ for convex and strongly convex loss functions respectively Hazan et al. (2016); Shalev-Shwartz et al. (2012). In recent years, due to the increasing volume of data, distributed online learning, especially decentralized methods, has attracted much attention. Examples of these works include Kamp et al. (2014); Shahrampour and Jadbabaie (2017); Lee et al. (2016). Notably, Zhao et al. (2019) shares a similar problem definition and theoretical result as our paper. However, single-sided communication is not allowed in their setting, restricting their results.

## 3 PROBLEM SETTING

In this paper, we consider federated learning with $n$ clients (a.k.a., nodes). Each client can be either an edge server or some other kind of computing device such as smart phone, which has local private data and the local machine learning model $\mathbf{x}_i$ stored on it. We assume the topological structure of the network of these $n$ nodes can be represented by a directed graph $\mathcal{G} = (\text{nodes} : [n], \text{edges} : E)$ with vertex set $[n] = \{1, 2, \ldots, n\}$ and edge set $E \subset [n] \times [n]$. If there exist an edge $(u, v) \in E$, it means node $u$ and node $v$ have network connection and $u$ can directly send messages to $v$.

Let $\mathbf{x}_t^{(i)}$ denote the local model on the $i$-th node at iteration $t$. In each iteration, node $i$ receives a new sample and computes a prediction for this new sample according to the current model $\mathbf{x}_t^{(i)}$ (e.g., it may recommend some items to the user in the online recommendation system). After that, a loss function, $f_{i,t}(\cdot)$ associated with that new sample is received by node $i$. The typical goal of online learning is to minimize the *regret*, which is defined as the difference between the summation of the losses incurred by the nodes' prediction and the corresponding loss of the global optimal model $\mathbf{x}^*$:

$$\tilde{\mathcal{R}}_T := \sum_{t=1}^{T} \sum_{i=1}^{n} \left( f_{i,t}(\mathbf{x}_t^{(i)}) - f_{i,t}(\mathbf{x}^*) \right),$$

where $\mathbf{x}^* = \arg\min_{\mathbf{x}} \sum_{t=1}^{T} \sum_{i=1}^{n} f_{i,t}(\mathbf{x})$ is the optimal solution.

However, here we consider a more general online setting: the loss function of the $i$-th node at iteration $t$ is $f_{i,t}(\cdot; \boldsymbol{\xi}_{i,t})$, which is additionally parametrized by a random variable $\boldsymbol{\xi}_{i,t}$. This $\boldsymbol{\xi}_{i,t}$ is drawn from the distribution $D_{i,t}$, and is mutually independent in terms of $i$ and $t$, and we call this part as the *stochastic* component of loss function $f_{i,t}(\cdot; \boldsymbol{\xi}_{i,t})$. The stochastic component can be utilized to characterize the internal randomness of nodes' data, and the potential connection among different nodes. For example, music preference may be impacted by popular trends on the Internet, which can be formulated by our model by letting $D_{i,t} \equiv D_t$ for all $i \in [n]$ with some time-varying distribution $D_t$. On the other hand, function $f_{i,t}(\cdot; \cdot)$ is the *adversarial* component of the loss, which may include, for example, user's profile, location, etc. Therefore, the objective regret naturally becomes the expectation of all the past losses:

$$\mathcal{R}_T := \mathbb{E}_{\Xi_{n,T} \sim \mathcal{D}_{n,T}} \left\{ \sum_{t=1}^{T} \sum_{i=1}^{n} \left( f_{i,t}(\mathbf{x}_t^{(i)}; \boldsymbol{\xi}_t^{(i)}) - f_{i,t}(\mathbf{x}^*; \boldsymbol{\xi}_t^{(i)}) \right) \right\} \tag{1}$$

with $\mathbf{x}^* = \arg\min_{\mathbf{x}} \mathbb{E}_{\Xi_{n,T} \sim \mathcal{D}_{n,T}} \sum_{t=1}^{T} \sum_{i=1}^{n} f_{i,t}(\mathbf{x}; \boldsymbol{\xi}_t^{(i)})$.

One benefit of the above formulation is that it partially resolves the non-I.I.D. issue in federated learning. A fundamental assumption in many traditional distributed machine learning methods is that the data samples stored on all nodes are I.I.D., which fails to hold for federated learning since the data on each user's device is highly correlated to that user's preferences and habits. However, our formulation does not require the I.I.D. assumption to hold for the adversarial component at all. Even though the random samples for the stochastic component still need to be independent, they are allowed to be drawn from different distributions.

Finally, one should note that online optimization also includes stochastic optimization (i.e., data samples are drawn from a fixed distribution) and offline optimization (i.e., data are already collected before optimization begins) as its typical cases Shalev-Shwartz et al. (2012). Hence, our setting covers a wide range of applications.

## 4 ONLINE PUSH-SUM ALGORITHM

In this section, we define the construction of the confusion matrix and introduce the proposed algorithm.

### 4.1 CONSTRUCTION OF CONFUSION MATRIX

One important parameter of the algorithm is the confusion matrix $\mathbf{W}$. $\mathbf{W}$ is a matrix depending on the network topology $\mathcal{G}$, which means $W_{ij} = 0$ if there is no directed edge $(i, j)$ in $\mathcal{G}$. If the value of $W_{ij}$ is large, the node $i$ will have a stronger impact on node $j$. However, $\mathbf{W}$ still allows flexibility where users can specify their weights associated with existing edges, meaning that even if there is a physical connection between two nodes, the nodes can decide against using the channel. For example, even if $(i, j) \in E$, user still can set $W_{ij} = 0$ if user $i$ thinks node $j$ is not trustworthy and therefore chooses to exclude the channel from $i$ to $j$.

Of course, there are still some constraints over $\mathbf{W}$. $\mathbf{W}$ must be a row stochastic matrix (i.e., each entry in $\mathbf{W}$ is non-negative, and the summation of each row is 1). This assumption is different from the one in classical decentralized distributed optimization, which typically assumes $\mathbf{W}$ is symmetric and doubly stochastic (e.g., Duchi et al. (2011)) (i.e., the summations of both rows and columns are all 1). Such a requirement is quite restrictive, because not all networks admit a doubly stochastic matrix (Gharesifard and Cortés (2010)), and relinquishing double stochasticity can introduce bias in optimization Ram et al. (2010); Tsianos and Rabbat (2012). As a comparison, our assumption that $\mathbf{W}$ is row stochastic will avoid such concerns since any non-negative matrix with at least one positive entry on each row (which is already implied by the connectivity of the graph) can be easily normalized into row stochastic. The relaxation of this assumption is crucial for federated learning, considering that the federated learning system usually involves complex network topology due to its large number of clients. Moreover, since each node only needs to make sure the summation of its out-weights is 1, there is no need for it to be aware of the global network topology, which significantly benefits the implementation of the federated learning system. Meanwhile, requiring $\mathbf{W}$ to be symmetric rules out the possibility of using asymmetric network topology and adopting sing-sided trust, while our method does not have such restriction.

### 4.2 ALGORITHM DESCRIPTION

The proposed online push-sum algorithm is presented in Algorithm 1. The algorithm design mainly follows the pattern of push-sum algorithm Tsianos et al. (2012), but here we further generalize it into the online setting.

The algorithm mainly consists of three steps:

1. Local update: each client $i$ applies the current local model $\mathbf{x}_t^{(i)}$ to obtain the loss function, based on which an intermediate local model $\mathbf{z}_{t+\frac{1}{2}}^{(i)}$ is computed;

2. Push: the weighted variable $W_{ij}\mathbf{z}_{t+\frac{1}{2}}^{(i)}$ is sent to $j$ for all its out neighbors $j$;

3. Sum: all the received $W_{ji}\mathbf{z}_{t+\frac{1}{2}}^{(j)}$ is summed and normalized to obtain the new model $\mathbf{x}_{t+1}^{(i)}$.

---

**Algorithm 1** Online Push-Sum (OPS) Algorithm

---

**Require:** Learning rate $\gamma$, number of iterations $T$, and the confusion matrix $\mathbf{W}$.

1: Initialize $\mathbf{x}_0^{(i)} = \mathbf{z}_0^{(i)} = \mathbf{0}$, $\omega_0^{(i)} = 1$ for all $i \in [n]$

2: **for** $t = 0, 1, ..., T-1$ **do**

3:     // For all users (say the $i$-th node $i \in [n]$)

4:     Apply local model $\mathbf{x}_t^{(i)}$ and suffer loss $f_{i,t}(\mathbf{x}_t^{(i)}; \boldsymbol{\xi}_t^{(i)})$

5:     Locally computes the intermedia variable

$$\mathbf{z}_{t+\frac{1}{2}}^{(i)} = \mathbf{z}_t^{(i)} - \gamma \nabla f_{i,t}\left(\mathbf{x}_t^{(i)}; \boldsymbol{\xi}_t^{(i)}\right)$$

6:     Send $\left(W_{ij}\mathbf{z}_{t+\frac{1}{2}}^{(i)}, W_{ij}\omega_t^{(i)}\right)$ to all $j \in \mathcal{N}_i^{\text{out}}$

7:     Update

$$\mathbf{z}_{t+1}^{(i)} = \sum_{k \in \mathcal{N}_i^{\text{in}}} W_{ki}\mathbf{z}_{t+\frac{1}{2}}^{(k)}$$

$$\omega_{t+1}^{(i)} = \sum_{k \in \mathcal{N}_i^{\text{in}}} W_{ki}\omega_t^{(k)}$$

$$\mathbf{x}_{t+1}^{(i)} = \frac{\mathbf{z}_{t+1}^{(i)}}{\omega_{t+1}^{(i)}}$$

8: **end for**

9: **return** $\mathbf{x}_T^{(i)}$ to node $i$

---

It should be noted an auxiliary variables $\mathbf{z}_{t+\frac{1}{2}}^{(i)}$ and $\mathbf{z}_{t+1}^{(i)}$ are used in the algorithm. Actually, they are used in the algorithm to clarify the description but may be easily removed in the practical implementation. Besides, another variable $\omega_{t+1}^{(i)}$ is also introduced, which is the normalizing factor of $\mathbf{z}_{t+1}^{(i)}$. $\omega_{t+1}^{(i)}$ plays an important role in the push-sum algorithm, since $\mathbf{W}$ is not doubly stochastic in our setting, and it is possible that the total weight $i$ receives does not equal to 1. The introduction of the normalizing factor $\omega_t^{(i)}$ helps the algorithm avoid issues brought by that $\mathbf{W}$ is not doubly stochastic. Furthermore, when $\mathbf{W}$ becomes doubly stochastic, it can be easily verified that $\omega_t^{(i)} \equiv 1$ and $\mathbf{x}_t^{(i)} \equiv \mathbf{z}_t^{(i)}$ for any $i$ and $t$, then Algorithm 1 reduces to the distributed online gradient method proposed by Zhao et al. (2019).

In the algorithm, the local data, which is encoded in the gradient $f_{i,t}(\mathbf{x}_t^{(i)}; \boldsymbol{\xi}_t)$ Shokri and Shmatikov (2015), is only utilized in updating local model. What neighboring nodes exchanges are only limited to the local models.

### 4.3 REGRET ANALYSIS

In this subsection, we provide regret bound analysis of OPS algorithm. Due to the limitation of space, the detail proof is deferred to the appendix. For convenience, we first denote

$$F_{i,t}(\mathbf{x}) := \mathop{\mathbb{E}}_{\boldsymbol{\xi}_{i,t} \sim D_{i,t}} f_{i,t}(\mathbf{x}; \boldsymbol{\xi}_{i,t}).$$

To carry out the analysis, the following assumptions are required:

**Assumption 1.** *We make the following assumptions throughout this paper: (1) The topological graph $\mathcal{G}$ is strongly connected; $\mathbf{W}$ is row stochastic; (2) For any $i \in [n]$ and $t \in [T]$, the loss function $f_{i,t}(\mathbf{x}; \boldsymbol{\xi}_{i,t})$ is convex in $\mathbf{x}$; (3) The problem domain is bounded such that for any two vectors $\mathbf{x}$ and $\boldsymbol{y}$ we always have $\|\mathbf{x} - \boldsymbol{y}\|^2 \leq R$; (4) The norm of the expected gradient $\nabla F_{i,t}(\cdot)$ is bounded, i.e., there exist constant $G > 0$ such that $\|\nabla F_{i,t}(\mathbf{x})\|^2 \leq G^2$ for any $i$, $t$ and $\mathbf{x}$; (5) The gradient variance is also bounded by $\sigma^2$, namely,*

$$\mathop{\mathbb{E}}_{\boldsymbol{\xi}_{i,t} \sim D_{i,t}} \|\nabla f_{i,t}(\mathbf{x}; \boldsymbol{\xi}_{i,t}) - \nabla F_{i,t}(\mathbf{x})\|^2 \leq \sigma^2.$$

Here constant $G$ provides an upper bound for the adversarial component. On the other hand, $\sigma$ measures the magnitude of stochasticity brought by the stochastic component. When $\sigma = 0$, the problem setting simply reduces back to normal distributed online learning. The strong connectivity assumption is necessary to ensure that the information can be exchanged between any two nodes. As for the convexity and the domain boundedness assumptions, they are quite common in online learning literature, such as Hazan et al. (2016).

Equipped with these assumptions, now we are ready to present the convergence result:

**Theorem 2.** *If we set*

$$\gamma = \frac{\sqrt{n}R}{\sigma\sqrt{1 + nC_2} + G\sqrt{nC_1T}}, \tag{2}$$

*the regret of OPS can be bounded by:*

$$\mathcal{R}_T \leq \mathcal{O}\left(nGR\sqrt{T} + \sigma R\left(1 + \sqrt{nC_2}\right)\sqrt{nT}\right), \tag{3}$$

*where $C_1$ and $C_2$ are two constants defined in the appendix.*

Note that when $n = 1$ and $\sigma = 0$, where the problem setting just reduces to normal online optimization, the implied regret bound $\mathcal{O}(GR\sqrt{T})$ exactly matches the lower bound of online optimization Hazan et al. (2016). Moreover, our result also matches the convergence rate of centralized online learning where $q = 0$ for fully connected networks. Hence, we can conclude that the OPS algorithm has optimal dependence on $T$.

This bound has a linear dependence on the number of nodes $n$, but it is easy to understand. First, we have defined the regret to be the summation of the losses on all the nodes. Increasing $n$ makes the regret naturally larger. Second, our federated learning setting is different from the typical distributed learning in that I.I.D. assumption does not hold here. Each node contains distinct local data that may be drawn from totally different distributions. Therefore, adding more nodes is not helpful for decreasing the regret of existing clients.

Moreover, we also prove that the difference of the model $\mathbf{x}_t^{(i)}$ on each worker could be bounded using the following theorem:

**Theorem 3.** *If we set $\gamma$ as (2), the difference of the model $\mathbf{x}_t^{(i)}$ on each worker admits a faster convergence rate than regret:*

$$\frac{1}{T}\sum_i^n \sum_{t=0}^T \left\|\mathbf{x}_{t+1}^{(i)} - \overline{\mathbf{z}}_{t+1}\right\|^2 \leq \mathcal{O}\left(\frac{nGR + nR\sigma}{T}\right).$$

Hence, the models on all clients' devices will finally converge to the same one with rate $\mathcal{O}(1/T)$.

### 4.4 PRIVACY PROTECTION

Our proposed algorithm has several advantages concerning privacy protection.

First, as we have mentioned, OPS runs in a decentralized way and exchanges models instead of gradients or training samples, which is already proven effective for reducing the risk of privacy leakage Bellet et al. (2017). Second, OPS runs in a decentralized and asymmetric fashion. These properties create difficulties for many attacking methods such as Nasr et al. (2018). In order to infer the data of other clients, the attacker needs to know the reactions of other nodes after the attack is injected, which is impossible when the connections are single-sided. Even though the attack will spread among the whole network and finally return to the attacker, it is still hard for the attacker to distinguish whether the information he receives from its neighbors is already affected by the attack or not, since he is unaware of the global topology.

## 5 EXPERIMENTS

We compare the performance of our proposed Online Push-Sum (OPS) method with that of Decentralized Online Gradient method (DOL) and Centralized Online Gradient method (COL), and then evaluate the effectiveness of OPS in different network size and network topology density settings.

### 5.1 IMPLEMENTATION AND SETTINGS

We consider online logistic regression with squared $\ell_2$ norm regularization:

$$f_{i,t}\left(\mathbf{x}; \boldsymbol{\xi}_{i,t}\right) = \log\left(1 + \exp\left(-\mathbf{y}_{i,t}\mathbf{A}_{i,t}^\top\mathbf{x}\right)\right) + \frac{\lambda}{2}\|\mathbf{x}\|^2,$$

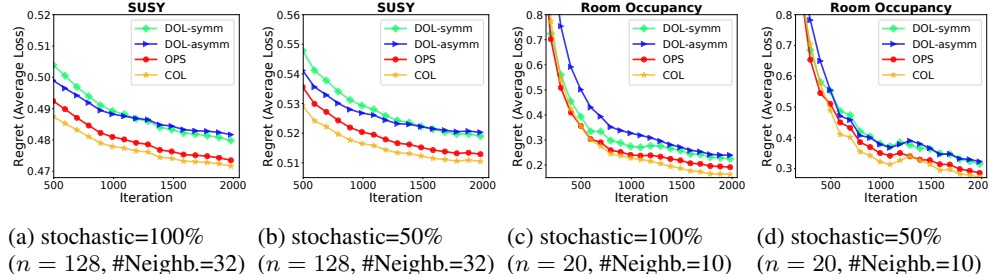

(a) stochastic=100%    (b) stochastic=50%    (c) stochastic=100%    (d) stochastic=50%
($n = 128$, #Neighb.=32)   ($n = 128$, #Neighb.=32)   ($n = 20$, #Neighb.=10)   ($n = 20$, #Neighb.=10)

Figure 2: Comparison among OPS, DOL (Decentralized Online Learning) and COL (Centralized Online Learning)

where regularization coefficient $\lambda$ is set to $10^{-4}$. $\boldsymbol{\xi}_{i,t}$ is the stochastic component of the function $f_{i,t}$ introduced in Section § 3, which is encoded in the random data sample $(\mathbf{A}_{i,t}, \mathbf{y}_{i,t})$. We evaluate the learning performance by measuring the average loss

$$\frac{1}{nT}\mathbb{E}_{\Xi_{n,T}}\sum_{i=1}^{n}\sum_{t=1}^{T}f_{i,t}\left(\mathbf{x}_{i,t}; \boldsymbol{\xi}_{i,t}\right),$$

instead of using the dynamic regret (1) directly, since the optimal reference point $x^*$ is the same for all the methods. The learning rate $\gamma$ in Algorithm 1 is tuned to be optimal for each dataset separately. The experiment implementation is based on Python 3.7.0, PyTorch 1.2.0, NetworkX 2.3, and scikit-learn 0.20.3. The source code along with other information concerning the experiment such as the setting of the hyper-parameters is provided in the supplementary materials.

**Dataset**   Experiments were run on two real-world public datasets: *SUSY*[1] and *Room-Occupancy*[2]. SUSY and *Room-Occupancy* are both large-scale binary classification datasets, containing 5,000,000 and 20,566 samples, respectively. Each dataset is split into two subsets: the stochastic data and the adversarial data. The stochastic data is generated by allocating a fraction of samples (e.g., 50% of the whole dataset) to nodes randomly and uniformly. The adversarial data is generated by conducting on the remaining dataset to produce $n$ clusters and then allocating every cluster to a node. As we analyzed previously, only the scattered stochastic data can boost the model performance by intra-node communication. For each node, this pre-acquired data is transformed into streaming data to simulate online learning.

## 5.2   COMPARISON WITH DOL AND COL

To compare OPS with DOL and COL, a network size with 128 nodes and 20 nodes are selected for SUSY and Room-Occupancy, respectively. For COL, its confusion matrix $\mathbf{W}$ is fully-connected (doubly stochastic matrix). For DOL and OPS, they are run with the same network topology and the same row stochastic matrix (asymmetric confusion matrix) to maintain a fair comparison. Such asymmetric confusion is constructed by setting each node's number of neighbors as a random value which is smaller than a fixed upper bound and also ensures the strong connectivity of the whole network (this upper-bound neighbor number is set to 32 for the SUSY dataset, while 10 is set for the Room-Occupancy dataset). Since DOL typically requires the network to be the symmetric and doubly stochastic confusion matrix, DOL is run in two settings for comparison. In the first setting, in order to meet the assumption of the symmetry and doubly stochasticity, all unidirectional connections are removed in the confusion matrix so that the row stochastic confusion matrix degenerates into a doubly stochastic matrix. This setting is labeled as *DOL-Symm* in Figure 2. In another setting, DOL is forced to run on the asymmetric network where each node naively aggregates its received models without considering whether its sending weights are equal to its receiving weights. *DOL-Asymm* is used to label this setting in Figure 2.

As illustrated in Figure 2, in both two datasets, OPS outperforms *DOL-Symm* in the row stochastic confusion matrix. This demonstrates that incorporating unidirectional communication can help to boost the model performance. In other words, OPS gains better performance in the single-sided

---

[1] https://www.csie.ntu.edu.tw/~cjlin/libsvmtools/datasets/binary.html#SUSY

[2] https://archive.ics.uci.edu/ml/datasets/Occupancy+Detection+

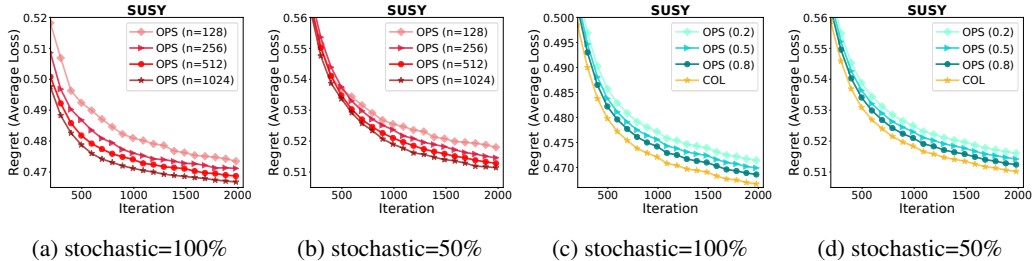

Figure 3: Evaluation on different network sizes and densities

trust network under the setting of federated learning. OPS also works better than *DOL-Asymm*. Although *DOL-Asymm* utilizes additional unidirectional connections, in some cases its performance is even worse than *DOL-Symm* (e.g., Figure 2a). This phenomenon is most likely attributed to its simple aggregation pattern, which causes decreased performance in *DOL-Asymm* when removing the doubly stochastic matrix assumption. These two observations confirm the effectiveness of OPS in a row stochastic confusion matrix, which is consistent with our theoretical analysis.

Comparing Figure 2c and Figure 2d, we also observe that when increasing the ratio of the stochastic component, the average loss (regret) becomes smaller. It is reasonable that OPS achieves slightly worse performance than COL because OPS works in a sparsely connected network where information exchanging is much less than COL. We use the COL as the baseline in all experiments.

Only the number of iterations instead of the actual running time is considered in the experiment. It is redundant to present the actual running time. Because the centralized method requires more time for each iteration due to the network congestion in the central node, OPS usually outperforms COL in terms of running time.

## 5.3 EVALUATION ON DIFFERENT NETWORK SIZES

Figure 3a and 3b summarizes the evaluation of OPS in different network sizes (in the *SUSY* dataset, 128, 256, 512, 1024 are set). The upper-bound neighbor number is aligned to the same value among different network sizes to isolate its impact. As we can see, in every dataset, the average loss (regret) curve in different network sizes is close on a small scale. These observations demonstrate OPS is robust to the network size. Furthermore, the average loss (regret) is smaller in larger network size (i.e., the curve of the $n = 1024$ network size is lower than others), which also demonstrates that more stochastic samples provided by more nodes can naturally accelerate the convergence. Due to limitation of space, the results on the other dataset is deferred to the appendix.

## 5.4 EVALUATION ON NETWORK DENSITY

We also evaluate the performance of OPS in different network densities. We fix the network size to 512 for *SUSY* dataset. Network density is defined as the ratio of the upper-bound random neighbor number per node to the size of the network (e.g., if the ratio is 0.5 in *SUSY*, it means 256 is set as the upper-bound neighbor number for each node). We can see from Figure 3c and 3d that as the network density increased, the average loss (regret) decreased. This observation also proves that our proposed OPS algorithm can work well in different network densities, and can gain more benefits from a denser row stochastic matrix. This benefit can also be understood intuitively: in a federated learning network, a user's model performance will improve if it communicates with more users. The results of *Room Occupancy* are also deferred to the appendix.

## 6 CONCLUSIONS

Decentralized federated learning with single-sided trust is a promising framework for solving a wide range of problems. In this paper, the online push-sum algorithm is developed for this setting, which is able to handle complex network topology and is proven to have an optimal convergence rate. The regret-based online problem formulation also extends its applications. We tested the proposed OPS algorithm in various experiments, which have empirically justified its efficiency.

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

## A    Proofs

**Notations:**    Below we use the following notation in our proof

- $\nabla F_t(\mathbf{X}_t) := \left[ \nabla F_{1,t}\left(\mathbf{x}_t^{(1)}\right), \cdots, \nabla F_{n,t}\left(\mathbf{x}_t^{(n)}\right) \right]$
- $\mathbf{X}_t := \left[ \mathbf{x}_t^{(1)}, \mathbf{x}_t^{(2)}, ..., \mathbf{x}_t^{(n)} \right]$
- $\mathbf{G}_t := \left[ \nabla f_{1,t}(\mathbf{x}_t^1; \boldsymbol{\xi}_t^1), \ldots, \nabla f_{n,t}(\mathbf{x}_t^n; \boldsymbol{\xi}_t^n) \right]$

Here we first present the proof Theorem 2, then we will present some key lemmas along with the proof of Theorem 3. The following theorem is the key to prove Theorem 2:

**Theorem 4.** *For the online push-sum algorithm with step size $\gamma > 0$, it holds that*

$$\mathcal{R}_T \leq G^2 T n \gamma C_1 + \sigma^2 T \gamma (1 + n C_2) + \frac{nR^2}{2\gamma}, \tag{4}$$

*where*

$$C_1 := \frac{8Cq}{\delta_{\min}(1-q)} + 1, \quad C_2 := \frac{2Cq}{\delta_{\min}(1-q)},$$

*and $C$, $q$ and $\delta_{min}$ are some constants defined in later lemmas.*

*Proof.* Since the loss function $f_{i,t}(\cdot)$ is assumed to be convex, which leads to

$$
\mathbb{E}_t \sum_{i=1}^n f_{i,t}\left(\mathbf{x}_t^{(i)}; \boldsymbol{\xi}_t^{(i)}\right) - nF_t(\mathbf{x}^*)
$$

$$
=\mathbb{E}_t \sum_{i=1}^n \left( f_{i,t}\left(\mathbf{x}_t^{(i)}; \boldsymbol{\xi}_t^{(i)}\right) - f_{i,t}\left(\mathbf{x}^*; \boldsymbol{\xi}_t^{(i)}\right) \right)
$$

$$
\leq \mathbb{E}_t \sum_{i=1}^n \left\langle \nabla f_{i,t}\left(\mathbf{x}_t^{(i)}; \boldsymbol{\xi}_t^{(i)}\right), \mathbf{x}_t^{(i)} - \mathbf{x}^* \right\rangle
$$

$$
= \underbrace{\mathbb{E}_t \sum_{i=1}^n \left\langle \nabla f_{i,t}\left(\mathbf{x}_t^{(i)}; \boldsymbol{\xi}_t^{(i)}\right), \mathbf{x}_t^{(i)} - \overline{\mathbf{z}}_t \right\rangle}_{:=I_{1t}} + \underbrace{\mathbb{E}_t \sum_{i=1}^n \left\langle \nabla f_{i,t}\left(\mathbf{x}_t^{(i)}; \boldsymbol{\xi}_t^{(i)}\right), \overline{\mathbf{z}}_t - \mathbf{x}^* \right\rangle}_{:=I_{2t}}.
$$

For $I_{2t}$, we have

$$
\mathbb{E}_t \sum_{i=1}^n \left\langle \nabla f_{i,t}\left(\mathbf{x}_t^{(i)}; \boldsymbol{\xi}_t^{(i)}\right), \overline{\mathbf{z}}_t - \mathbf{x}^* \right\rangle
$$

$$
= \frac{n}{\gamma} \mathbb{E}_t \left\langle \frac{\gamma}{n} \sum_{i=1}^n \nabla f_{i,t}\left(\mathbf{x}_t^{(i)}; \boldsymbol{\xi}_t^{(i)}\right), \overline{\mathbf{z}}_t - \mathbf{x}^* \right\rangle
$$

$$
= \frac{n}{2\gamma} \mathbb{E}_t \left( \left\| \frac{\gamma}{n} \sum_{i=1}^n \nabla f_{i,t}\left(\mathbf{x}_t^{(i)}; \boldsymbol{\xi}_t^{(i)}\right) \right\|^2 + \|\overline{\mathbf{z}}_t - \mathbf{x}^*\|^2 - \left\| \overline{\mathbf{z}}_t - \mathbf{x}^* - \frac{\gamma}{n} \sum_{i=1}^n \nabla f_{i,t}\left(\mathbf{x}_t^{(i)}; \boldsymbol{\xi}_t^{(i)}\right) \right\|^2 \right)
$$

$$
= \frac{n}{2\gamma} \mathbb{E}_t \left( \left\| \frac{\gamma}{n} \sum_{i=1}^n \nabla f_{i,t}\left(\mathbf{x}_t^{(i)}; \boldsymbol{\xi}_t^{(i)}\right) \right\|^2 + \|\overline{\mathbf{z}}_t - \mathbf{x}^*\|^2 - \|\overline{\mathbf{z}}_{t+1} - \mathbf{x}^*\|^2 \right)
$$

$$
\leq \frac{n}{2\gamma} \mathbb{E}_t \left( \gamma^2 G^2 + \frac{\gamma^2 \sigma^2}{n} + \|\overline{\mathbf{z}}_t - \mathbf{x}^*\|^2 - \|\overline{\mathbf{z}}_{t+1} - \mathbf{x}^*\|^2 \right)
$$

Notice that for COL, we have $I_{1t} = 0$ because $\mathbf{x}_t^{(i)} = \overline{\mathbf{z}}_t$. So for DOL, in order to bound $I_{1t}$, we need to bound the difference $\left\| \mathbf{x}_t^{(i)} - \overline{\mathbf{z}}_t \right\|$ (using Lemma 8).

$$
\mathbb{E}_t \sum_{i=1}^n \left\langle \nabla f_{i,t}\left(\mathbf{x}_t^{(i)}; \boldsymbol{\xi}_t^{(i)}\right), \mathbf{x}_t^{(i)} - \overline{\mathbf{z}}_t \right\rangle
$$

$$
= \mathbb{E}_t \sum_{i=1}^n \left\langle \nabla F_{i,t}(\mathbf{x}_t^{(i)}), \mathbf{x}_t^{(i)} - \overline{\mathbf{z}}_t \right\rangle
$$

$$
\leq \mathbb{E}_t \sum_{i=1}^n \left( \alpha \left\| \nabla F_{i,t}\left(\mathbf{x}_t^{(i)}\right) \right\|^2 + \frac{1}{\alpha} \left\| \mathbf{x}_t^{(i)} - \overline{\mathbf{z}}_t \right\|^2 \right).
$$

Summing up the inequality above from $t = 1$ to $t = T$, we get

$$\sum_{t=1}^{T} \mathbb{E}_t \sum_{i=1}^{n} \left\langle \nabla f_{i,t}\left(\mathbf{x}_t^{(i)}; \boldsymbol{\xi}_t^{(i)}\right), \mathbf{x}_t^{(i)} - \overline{\mathbf{z}}_t \right\rangle$$

$$= \sum_{t=1}^{T} \mathbb{E}_t \sum_{i=1}^{n} \left\langle \nabla F_{i,t}\left(\mathbf{x}_t^{(i)}\right), \mathbf{x}_t^{(i)} - \overline{\mathbf{z}}_t \right\rangle$$

$$\leq \sum_{t=1}^{T} \mathbb{E}_t \sum_{i=1}^{n} \left( \alpha \left\| \nabla F_{i,t}\left(\mathbf{x}_t^{(i)}\right) \right\|^2 + \frac{1}{\alpha} \left\| \mathbf{x}_t^{(i)} - \overline{\mathbf{z}}_t \right\|^2 \right)$$

$$= \sum_{t=1}^{T} \left( \alpha \mathbb{E}_t \left\| \nabla F_t(\mathbf{X}_t) \right\|_F^2 + \frac{1}{\alpha} \mathbb{E}_t \left\| \mathbf{X}_t - \overline{\mathbf{z}}_t \right\|_F^2 \right)$$

$$\leq \alpha \sum_{t=1}^{T} \mathbb{E}_t \left\| \nabla F_t\left(\mathbf{X}_t\right) \right\|_F^2 + \frac{4\gamma^2 C^2 q^2}{\alpha \delta_{\min}^2 (1-q)^2} \sum_{t=1}^{T} \mathbb{E}_t \left\| \mathbf{G}_t \right\|_F^2$$

$$\leq \alpha \sum_{t=1}^{T} \mathbb{E}_t \left\| \nabla F_t\left(\mathbf{X}_t\right) \right\|_F^2 + \frac{4\gamma^2 C^2 q^2}{\alpha \delta_{\min}^2 (1-q)^2} \sum_{t=1}^{T} \left( \mathbb{E}_t \left\| \nabla F_t(\mathbf{X}_t) \right\|_F^2 + n\sigma^2 \right).$$

Choosing $\alpha = \frac{2\gamma C q}{\delta_{\min}(1-q)}$, we have

$$\sum_{t=1}^{T} \mathbb{E}_t \sum_{i=1}^{n} \left\langle \nabla f_{i,t}\left(\mathbf{x}_t^{(i)}; \boldsymbol{\xi}_t^{(i)}\right), \mathbf{x}_t^{(i)} - \overline{\mathbf{z}}_t \right\rangle \leq \frac{8n\gamma C T q G^2}{\delta_{\min}(1-q)} + \frac{2n\gamma C q \sigma^2 T}{\delta_{\min}(1-q)}$$

So we have

$$\sum_{t=1}^{T} \mathbb{E}_t \sum_{i=1}^{n} f_{i,t}\left(\mathbf{z}_t^{(i)}; \boldsymbol{\xi}_t^{(i)}\right) - nF(\mathbf{x}^*)$$

$$\leq \frac{8n\gamma C T q G^2}{\delta_{\min}(1-q)} + \frac{2\gamma C q \sigma^2 T}{\delta_{\min}(1-q)} + \frac{n}{2n\gamma} \sum_{t=1}^{T} \left( \gamma^2 G^2 + \frac{\gamma^2 \sigma^2}{n} + \mathbb{E}_t \left\| \overline{\mathbf{z}}_t - \mathbf{x}^* \right\|^2 - \mathbb{E}_t \left\| \overline{\mathbf{z}}_{t+1} - \mathbf{x}^* \right\|^2 \right)$$

$$\leq G^2 T n\gamma \left( \frac{8Cq}{\delta_{\min}(1-q)} + 1 \right) + \sigma^2 T \gamma \left( 1 + \frac{2nCq}{\delta_{\min}(1-q)} \right) + \frac{n}{2\gamma} \sum_{t=1}^{T} \left( \mathbb{E}_t \left\| \overline{\mathbf{z}}_t - \mathbf{x}^* \right\|^2 - \mathbb{E}_t \left\| \overline{\mathbf{z}}_{t+1} - \mathbf{x}^* \right\|^2 \right)$$

$$\leq G^2 T n\gamma \left( \frac{8Cq}{\delta_{\min}(1-q)} + 1 \right) + \sigma^2 T \gamma \left( 1 + \frac{2nCq}{\delta_{\min}(1-q)} \right) + \frac{nR^2}{2\gamma}$$

$$= C_1 n G^2 T \gamma + (1 + nC_2)\sigma^2 T \gamma + \frac{nR^2}{2\gamma}.$$

Notice that Theorem 2 can be easily verified by setting $\gamma = \frac{\sqrt{n}R}{\sqrt{(1+nC_2)\sigma^2 + \sqrt{nC_1 G^2 T}}}$. $\qquad \square$

Next, we will present two lemmas for our proof of Lemma 8. The proofs of following two lemmas can be found in existing literature Nedić and Olshevsky (2014; 2016); Assran and Rabbat (2018); Assran et al. (2018).

**Lemma 5.** *Under the Assumption 1, there exists a constant $\delta_{\min} > 0$ such that for any t, the following holds*

$$\sum_{j=1}^{n} [\mathbf{W}^{t\top} \mathbf{W}^{t\top} ... \mathbf{W}^{0\top}]_{ij} \geq \delta_{\min} \geq \frac{1}{n^n}, \ \forall i \tag{5}$$

*where $\mathbf{W}^t$ is a row stochastic matrix.*

**Lemma 6.** *Under the Assumption 1, for any t, there always exists a stochastic vector $\psi(t)$ and two constants $C = 4$ and $q = 1 - n^{-n} < 1$ such that for any $s$ satisfying $s \leq t$, the following inequality holds*

$$\left| [\mathbf{W}^{t\top} \mathbf{W}^{t\top} \cdots \mathbf{W}^{s+1\top} \mathbf{W}^{s\top}]_{ij} - \psi_i(t) \right| \leq Cq^{t-s}, \forall i, j$$

*where $\mathbf{W}^t$ is a row stochastic matrix, and $\psi(t)$ is a vector with $\psi_i(t)$ being its $i$-th entry.*

**Lemma 7.** *Given two non-negative sequences $\{a_t\}_{t=1}^{\infty}$ and $\{b_t\}_{t=1}^{\infty}$ that satisfying*

$$a_t = \sum_{s=1}^{t} \rho^{t-s} b_s, \tag{6}$$

*with $\rho \in [0, 1)$, we have*

$$D_k := \sum_{t=1}^{k} a_t^2 \leq \frac{1}{(1-\rho)^2} \sum_{s=1}^{k} b_s^2.$$

*Proof.* From the definition, we have

$$S_k = \sum_{t=1}^{k} \sum_{s=1}^{t} \rho^{t-s} b_s = \sum_{s=1}^{k} \sum_{t=s}^{k} \rho^{t-s} b_s = \sum_{s=1}^{k} \sum_{t=0}^{k-s} \rho^t b_s \leq \sum_{s=1}^{k} \frac{b_s}{1-\rho}, \tag{7}$$

$$D_k = \sum_{t=1}^{k} \sum_{s=1}^{t} \rho^{t-s} b_s \sum_{r=1}^{t} \rho^{t-r} b_r$$

$$= \sum_{t=1}^{k} \sum_{s=1}^{t} \sum_{r=1}^{t} \rho^{2t-s-r} b_s b_r$$

$$\leq \sum_{t=1}^{k} \sum_{s=1}^{t} \sum_{r=1}^{t} \rho^{2t-s-r} \frac{b_s^2 + b_r^2}{2}$$

$$= \sum_{t=1}^{k} \sum_{s=1}^{t} \sum_{r=1}^{t} \rho^{2t-s-r} b_s^2$$

$$\leq \frac{1}{1-\rho} \sum_{t=1}^{k} \sum_{s=1}^{t} \rho^{t-s} b_s^2$$

$$\leq \frac{1}{(1-\rho)^2} \sum_{s=1}^{k} b_s^2.$$

$\square$

Based on the above three lemmas, we can obtain the following lemma.

**Lemma 8.** *Under the Assumption 1, the updating rule of Algorithm 1 leads to the following inequality*

$$\sum_{i}^{n} \sum_{t=0}^{T} \left\| \mathbf{x}_{t+1}^{(i)} - \bar{\mathbf{z}}_{t+1} \right\|_2^2 \leq \frac{4\gamma^2 C^2 q^2}{\delta_{\min}^2 (1-q)^2} \sum_{s=0}^{t} \|\mathbf{G}_s\|_F^2,$$

*where $\gamma$ is the step size, and $C = 4, \delta_{\min} \geq n^{-n}, q = 1 - n^{-n}$ are constants. $\mathbf{G}_s$ is the matrix for the stochastic gradient at time $s$ (e.g., the $i$-th column is the stochastic gradient vector on node $i$ at time $s$).*

*Proof.* The updating rule of OPS can be formulated as

$$\mathbf{Z}_{t+1} = (\mathbf{Z}_t - \gamma \mathbf{G}_t) \mathbf{W}$$

$$\omega_{t+1} = \mathbf{W}^\top \omega_t$$

$$\mathbf{X}_{t+1} = \mathbf{Z}_{t+1} [\text{diag}(\omega_{t+1})]^{-1}$$

where $\mathbf{W}$ is a row stochastic matrix. $\mathbf{X}_t = [\mathbf{x}_t^{(1)}, \mathbf{x}_t^{(2)}, ..., \mathbf{x}_t^{(n)}]$ is a matrix whose each column is $\mathbf{x}_t^{(i)}$. $G_t$ is the matrix of gradient, whose each column is the stochastic gradient at $\mathbf{z}_t^{(i)}$ on node $i$. $\mathbf{Z}_t = [\mathbf{z}_t^{(1)}, ..., \mathbf{z}_t^{(n)}]$ is the matrix whose each column is $\mathbf{z}_t^{(i)}$.

Assuming $X_0 = O$ and $\omega_0 = 1$, then we have

$$\mathbf{Z}_{t+1} = (\mathbf{Z}_t - \gamma \mathbf{G}_t)\mathbf{W} = ... = -\gamma \sum_{s=0}^{t} \mathbf{G}_s \mathbf{W}^{t-s+1}, \tag{8}$$

$$\overline{\mathbf{z}}_{t+1} = \overline{\mathbf{z}}_t - \gamma \overline{\mathbf{g}}_t = ... = -\sum_{s=0}^{t} \gamma \overline{\mathbf{g}}_s, \tag{9}$$

$$\omega_{t+1} = \mathbf{W}^{t+1\top}\omega_0, \tag{10}$$

where $\overline{\mathbf{x}}_t = \mathbf{X}_t 1$ is the average of all variables on the $n$ nodes, and $\overline{\mathbf{g}}_t = \mathbf{G}_t 1$ is the averaged gradient. We have $\mathbf{W}1 = 1$ since $\mathbf{W}$ is a row stochastic matrix.

For $\omega_{t+1}$, according to Lemma 6, we decompose it as follows

$$\omega_{t+1} = \mathbf{W}^{t+1\top}\omega_0 = [\mathbf{W}^{t+1\top} - \psi(t)1^\top]\omega_0 + \psi(t)1^\top\omega_0 = [\mathbf{W}^{t+1\top} - \psi(t)1^\top]1 + n\psi(t), \tag{11}$$

since $\omega_0 = 1$.

On the other hand, according to Lemma 5, we also have

$$\omega_{t+1}^{(i)} = [\mathbf{W}^{t+1\top}1]^\top \mathbf{e}_i = \sum_{j=1}^{n} [\mathbf{W}^{t+1\top}]_{ij} \geq n\delta_{\min}, \tag{12}$$

where $\mathbf{e}_i$ is a vector with only the $i$-th entry being 1 and 0 for others.

We need to further bound the following term

$$\left\| \mathbf{x}_{t+1}^{(i)} - \overline{\mathbf{z}}_{t+1} \right\| = \gamma \left\| \frac{\mathbf{z}_{t+1}^{(i)}}{\omega_{t+1}^{(i)}} - \overline{\mathbf{z}}_{t+1} \right\|$$

$$= \gamma \left\| \sum_{s=0}^{t} \left( \frac{\mathbf{G}_s \mathbf{W}^{t-s+1}\mathbf{e}_i}{1^\top \mathbf{W}^{t+1}\mathbf{e}_i} - \frac{\mathbf{G}_s 1}{n} \right) \right\|$$

$$= \gamma \left\| \sum_{s=0}^{t} \frac{n\mathbf{G}_s \mathbf{W}^{t-s+1}\mathbf{e}_i - \mathbf{G}_s 11^\top \mathbf{W}^{t+1}\mathbf{e}_i}{n\omega_{t+1}^{(i)}} \right\|,$$

where the second equality is by (8), (9), and (10). We turn to bound the following term

$$\left\| \sum_{s=0}^{t} \frac{n\mathbf{G}_s \mathbf{W}^{t-s+1}\mathbf{e}_i - \mathbf{G}_s 11^\top \mathbf{W}^{t+1}\mathbf{e}_i}{n\omega_{t+1}^{(i)}} \right\|$$

$$\leq \frac{1}{n^2\delta_{\min}} \left\| \sum_{s=0}^{t} \left( n\mathbf{G}_s \mathbf{W}^{t-s+1}\mathbf{e}_i - \mathbf{G}_s 11^\top \mathbf{W}^{t+1}\mathbf{e}_i \right) \right\|,$$

where the first inequality is accordng to (12). Therefore, combining the results above, we can have

$$\sum_{i=1}^{n} \left\| \mathbf{x}_{t+1}^{(i)} - \overline{\mathbf{z}}_{t+1} \right\|_2^2 \leq \frac{\gamma^2}{n^4\delta_{\min}^2} \sum_{i=1}^{n} \left\| \sum_{s=0}^{t} \left( n\mathbf{G}_s \mathbf{W}^{t-s+1}\mathbf{e}_i - \mathbf{G}_s 11^\top \mathbf{W}^{t+1}\mathbf{e}_i \right) \right\|_2^2$$

$$\leq \frac{\gamma^2}{n^4\delta_{\min}^2} \left\| \sum_{s=0}^{t} \left( n\mathbf{G}_s \mathbf{W}^{t-s+1} - \mathbf{G}_s 11^\top \mathbf{W}^{t+1} \right) \right\|_F^2$$

where the second inequality is due to $\sum_{i=1}^{n} \|\mathbf{A}\mathbf{e}_i\|_2^2 = \|\mathbf{A}\|_F^2$.

It remains to bound the following term

$$
\left\| \sum_{s=0}^{t} \left( n\mathbf{G}_s \mathbf{W}^{t-s+1} - \mathbf{G}_s 11^\top \mathbf{W}^{t+1} \right) \right\|_F^2
$$

$$
= \left\| \sum_{s=0}^{t} \left( n\mathbf{G}_s \mathbf{W}^{t-s+1} - \mathbf{G}_s 1[1^\top (\mathbf{W}^{t+1} - \psi(t)1^\top)^\top + n\psi(t)^\top] \right) \right\|_F^2
$$

$$
= \left\| \sum_{s=0}^{t} \left( n\mathbf{G}_s[\mathbf{W}^{t-s+1} - 1\psi(t)^\top] - \mathbf{G}_s 11^\top[\mathbf{W}^{t+1} - 1\psi(t)^\top] \right) \right\|_F^2
$$

$$
\leq \left( \sum_{s=0}^{t} \left\| n\mathbf{G}_s[\mathbf{W}^{t-s+1} - 1\psi(t)^\top] \right\|_F + \sum_{s=0}^{t} \left\| \mathbf{G}_s 11^\top[\mathbf{W}^{t+1} - 1\psi(t)^\top] \right\|_F \right)^2
$$

$$
\leq \left( n \sum_{s=0}^{t} \|\mathbf{G}_s\|_F \|[\mathbf{W}^{t-s+1} - 1\psi(t)^\top]\|_F + \sum_{s=0}^{t} \|\mathbf{G}_s\|_F \|11^\top\|_F \|[\mathbf{W}^{t+1} - 1\psi(t)^\top]\|_F \right)^2
$$

$$
\leq n^2 \left( \sum_{s=0}^{t} \|\mathbf{G}_s\|_F \|[\mathbf{W}^{t-s+1} - 1\psi(t)^\top]\|_F + \sum_{s=0}^{t} \|\mathbf{G}_s\|_F \|[\mathbf{W}^{t+1} - 1\psi(t)^\top]\|_F \right)^2
$$

$$
\leq n^2 \left( \sum_{s=0}^{t} nCq^{t-s+1} \|\mathbf{G}_s\|_F + \sum_{s=0}^{t} nCq^{t+1} \|\mathbf{G}_s\|_F \right)^2
$$

$$
\leq 4n^4 C^2 q^2 \left( \sum_{s=0}^{t} q^{t-s} \|\mathbf{G}_s\|_F \right)^2
$$

where the third inequality is due to $\|11^\top\|_F = n$ and the fourth inequality is by Lemma 6 and the fact that $\|\mathbf{A}\|_F \leq n \cdot \max_{i,j} |A_{ij}|$ if $\mathbf{A} \in \mathbb{R}^{n \times n}$.

Therefore, if we combining all the above inequalities together, we can obtain

$$
\sum_{i=1}^{n} \left\| \mathbf{x}_{t+1}^{(i)} - \overline{\mathbf{z}}_{t+1} \right\|_2^2 \leq \frac{4\gamma^2 C^2 q^2}{\delta_{\min}^2} \left( \sum_{s=0}^{t} q^{t-s} \|\mathbf{G}_s\|_F \right)^2.
$$

Using Lemma 7, we have

$$
\sum_{t=0}^{T} \left( \sum_{s=0}^{t} q^{t-s} \|\mathbf{G}_s\|_F \right)^2 \leq \frac{1}{(1-q)^2} \sum_{t=0}^{T} \|\mathbf{G}_t\|_F^2,
$$

which leads to

$$
\sum_{t=0}^{T} \sum_{i=1}^{n} \left\| \mathbf{x}_{t+1}^{(i)} - \overline{\mathbf{z}}_{t+1} \right\|_2^2 \leq \frac{4\gamma^2 C^2 q^2}{\delta_{\min}^2 (1-q)^2} \sum_{t=0}^{T} \|\mathbf{G}_t\|_F^2,
$$

which completes the proof. $\qquad\square$

Actually, Theorem 3 is a corollary of Lemma 8 by setting $\gamma$ as the appropriate value.

## B    EXTRA EXPERIMENT RESULTS

### B.1    EVALUATION ON *Room Occupancy* DATASET

Due to the limitation of space, we only present the experiment results on *SUSY* dataset in Section 5.3 and 5.4. Related presents on *Room Occupancy* is shown in Figure 4 and Figure 5.

In Figure 4, we vary the number of clients in the network, from 6 to 20. In Figure 5, the network density is varied. All the results are consistent with the ones on *SUSY*.

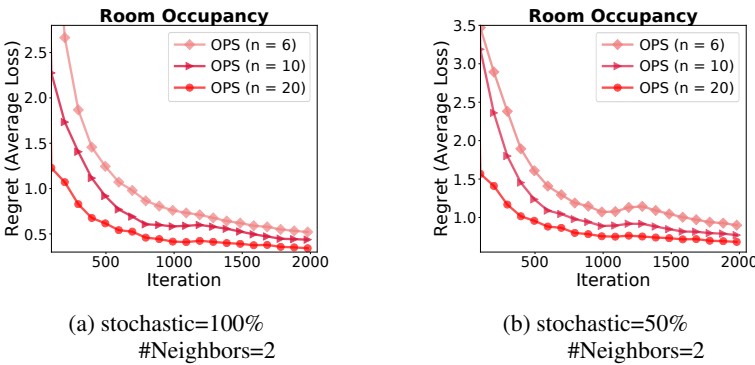

Figure 4: Evaluation on the Network Sizes

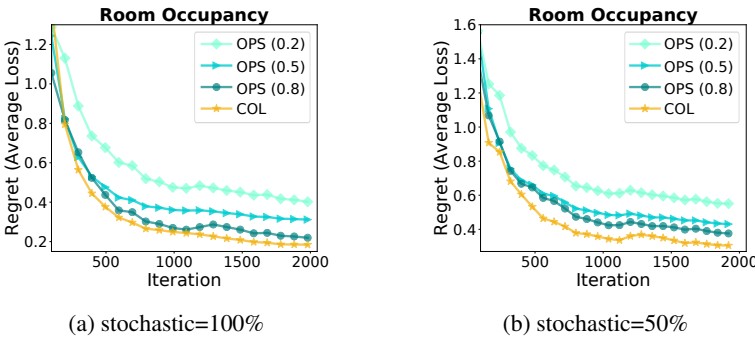

Figure 5: Evaluation on the Network Density

## B.2 COMPARISON WITH LOCAL ONLINE GRADIENT DESCENT

To justify the necessity of communication, we also compare OPS with the local online gradient descent (local OGD), where every node trains a local model without communicating with others. We run experiments in different ratios of the adversary and stochastic components based on settings in Figure 2. As we can see in Figure 6, we empirically prove that communication does have benefits in reducing regret. Moreover, as the ratio of the stochastic components increased, the regret of OPS decreases further. This also empirically proves that the stochastic component can benefit from the communication while the adversarial component does not.

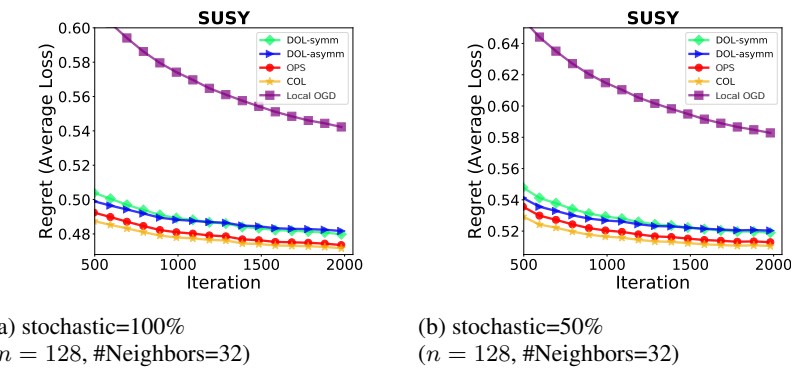

Figure 6: Comparison between OPS and Local OGD.

