# OpenReview forum: "Central Server Free Federated Learning over Single-sided Trust Social Networks"
_ICLR.cc/2021/Conference — Reject_

### Official Review · AnonReviewer2 · 2020-10-24
**The studied problem of decentralized federated learning in the context of social network seems to be interesting, but I think the novelty is low.**

**Rating:** 4
**Confidence:** 3

**Review:**

**Paper Summary:** This paper proposes a decentralized federated learning algorithm called Online Push-Sum (OPS) for peer-to-peer learning in the context of social networks with the property of single-sided trust .

**Questions for the authors**
1. Sec 1, it is mentioned that “Only models rather than local gradients are exchanged among clients in our algorithm.” My understanding is in most of the federated learning algorithms, either model parameters or difference of model parameters are exchanged (also noise can be added on top of them to guarantee differential privacy). So, it would be great if the authors further explain why this is a feature of their algorithm worth highlighting.
2. Maybe I have missed it, but in Sec 4.1, why do we need an extra parameter (i.e., $w_{t+1}^{i}$) for normalization? Why can't we just use the summation of the weights (i.e. $W_{ki}$) to do the normalization ? Can you explain further?
3. Sec 4.3, one assumption of the proposed algorithm is that the graph is strongly connected. I am afraid this may not be the case in practical applications. Fully decentralized algorithms for learning should be robust to the limited availability of the clients/nodes (with clients temporarily unavailable, dropping out or joining during the execution) and limited reliability of the network (with possible message drops). Interested to see the authors’ thoughts on this.

**Novelty**
The paper to me seems an incremental extension of the previous work (Zhao et al., 2019), and I think the novelty is a little thin.

**Areas to Improve**
I think it would be good to compare the proposed method with other existing Federated Learning methods such as (Dinh et al. NeurIPS, 2020) as well.

**Minor Concerns**
Page 2. Notation section. “denoting the sets of in neighbors of and out neighbors” -> "denoting the sets of out neighbors of and in neighbors" ?

**References**
1. Personalized Federated Learning with Moreau Envelopes (Dinh et al. NeurIPS, 2020)
2. Peer-to-peer federated learning on graphs (Lalitha et al. 2019). This is a relevant paper which is missing.

---

> ### Author Response · Authors · 2020-11-22
> **Response to Reviewer 2's comments**
>
> Thanks for your reviews of our paper. Below are our responses to your comments:
>
> ### Comment 1:
>
> > Sec 1, it is mentioned that “Only models rather than local gradients are exchanged among clients in our algorithm.” My understanding is in most of the federated learning algorithms, either model parameters or difference of model parameters are exchanged (also noise can be added on top of them to guarantee differential privacy). So, it would be great if the authors further explain why this is a feature of their algorithm worth highlighting.
>
> **Response:** Actually, this feature is the reason why we classify our algorithm as a federated learning method, instead of a feature distinguishing our method from other federated learning methods.
>
>
>
> ### Comment 2:
>
> > Maybe I have missed it, but in Sec 4.1, why do we need an extra parameter (i.e., $w_i^{t+1}$) for normalization? Why can't we just use the summation of the weights (i.e. $W_{ki}$) to do the normalization ? Can you explain further?
>
> **Response:** This is one of the advantages of our algorithm. Typically decentralized methods directly use the summation of the weights to do normalization. However, doing so would require the confusion matrix W to be doubly stochastic. We have clarified in Section 4.1 that such assumption is quite restrictive, e.g., each node needs to know the global topology . Instead, our algorithm removed such restrictions by the introduction of the dynamic weights $w_i^{t+1}$. Intuitively speaking, we gradually learn a suitable set of weights by iterating over these variables.
>
>
>
> ### Comment 3:
>
> > Sec 4.3, one assumption of the proposed algorithm is that the graph is strongly connected. I am afraid this may not be the case in practical applications. Fully decentralized algorithms for learning should be robust to the limited availability of the clients/nodes (with clients temporarily unavailable, dropping out or joining during the execution) and limited reliability of the network (with possible message drops). Interested to see the authors’ thoughts on this.
>
> **Response:** Yes, this is a very good point. We have already noticed this issue, and are planning to carry out the analysis on the dynamic graph, which allows clients to drop in and out. But in this paper, we still focus on the simple static graph case but single-sided topology is supported, which is a feature that has never been discussed by previous FL related works.
>
>
>
> ### Comment 4:
>
> > The paper to me seems an incremental extension of the previous work (Zhao et al., 2019), and I think the novelty is a little thin.
>
> **Response:** We politely disagree with this. We incorporated decentralized federated learning with this work, which results in a new method that can handle many real applications that (Zhao et al., 2019) can not. Besides, the combination of these two techniques is not trivial, and many technical challenges exist in the convergence analysis.
>
>
>
> ### Comment 5:
>
> > I think it would be good to compare the proposed method with other existing Federated Learning methods such as (Dinh et al. NeurIPS, 2020) as well.
>
>
> **Response:** According to the accepted paper list in NeurIPS 2020, there are two papers published by Dinh et al (https://papers.nips.cc/paper/2020/hash/1959eb9d5a0f7ebc58ebde81d5df400d-Abstract.html, and https://papers.nips.cc/paper/2020/hash/f4f1f13c8289ac1b1ee0ff176b56fc60-Abstract.html). Sorry, we are not sure which paper you refer, but we will discuss them in our revision.

---

> > ### Comment · AnonReviewer2 · 2020-11-24
> > **Unfortunately, I am not satisfied with the authors' responses.**
> >
> > Given the reviews of other reviewers and given that my comments (3,4, and 5) have not been addressed , unfortunately I have lowered my score. BTW, regarding the comment 5, I meant the second paper (i.e., Personalized Federated Learning with Moreau Envelopes). I am not sure if the first paper the authors cited has anything to do with FL.

---

### Official Review · AnonReviewer1 · 2020-10-28
**Interesting paper, but may not be novel**

**Rating:** 5
**Confidence:** 3

**Review:**

##########################################################################

Summary:

The paper proposed Online Push-Sum (OPS) method, which aims at solving decentralized federated optimization problems under a social network scenario where the centralized authority does not exist in a federated learning (FL) system. A social network application scenario is assumed by OPS where the graph is of single-sided trust. The author further extends the proposed OPS method to the online setting and provide regret analysis. The experimental study indicates that OPS is effective and converges faster than other decentralized online methods.

##########################################################################

Reasons for score:

Overall, I think the current manuscript is marginally below the acceptance threshold of ICLR conference. Studying the effectiveness of the central server free federated learning algorithm is a promising direction. The proposed algorithm is interesting and is with theoretical guarantees. However, the major concern is that the problem setting may not be novel enough. Moreover, the experimental justification of this paper can be improved.

##########################################################################

Pros:

1. The paper is well written. The research direction on studying central server free federated learning algorithms is promising.

2. The formulation and theoretical analysis of the proposed OPS method looks promising.

3. Experimental results under simulated federated learning under the social networking environment are provided to show the effectiveness of OPS.

##########################################################################

Cons:

1. The major concern on the proposed OPS method is its novelty. A series of central server-free federated learning algorithms have already been developed e.g. [1] and it seems the main contribution of OPS is to study one specific setting e.g. decentralized FL under the single-sided trust social network graph. Thus, the authors are expected to show either that OPS outperforms the previously proposed methods or its capability to handle a completely new area.
2. OPS seems to follow the Push-Sum algorithm [2], but is also extended to the online setting. But it is not quite clear why the online setting is important in the decentralized FL scenario.
3. The experimental justification of the proposed method is limited to linear models. However, most of the modern machine learning tasks are running over more complex models e.g. neural networks. The authors are highly encouraged to extend the scale of the experiments.

[1] https://arxiv.org/abs/1905.09435
[2] https://ieeexplore.ieee.org/document/6426375


#########################################################################

Minor Comments:
1. Missing references: [1-2].

[1] https://arxiv.org/abs/1912.04977
[2] https://arxiv.org/abs/1908.07873

---

> ### Author Response · Authors · 2020-11-22
> **Response to Reviewer 1's comments**
>
> Thanks for your reviews of our paper. Below are our responses to your comments:
>
> ### Comment 1:
>
> > The major concern on the proposed OPS method is its novelty. A series of central server-free federated learning algorithms have already been developed e.g. [1] and it seems the main contribution of OPS is to study one specific setting e.g. decentralized FL under the single-sided trust social network graph. Thus, the authors are expected to show either that OPS outperforms the previously proposed methods or its capability to handle a completely new area.
>
> **Response:** First, existing works have already proven that decentralized methods are superior than centralized methods in some applications (e.g., see Lian et al. (2017).).  Hence, we believe it is meaningful to study decentralized methods in FL settings. Second, as far as we know, decentralized FL in online settings is not previously studied. But we believe this setting is worthy of studying. Please see the next answer for detailed reasons.
>
>
>
> ### Comment 2:
>
> > OPS seems to follow the Push-Sum algorithm [2], but is also extended to the online setting. But it is not quite clear why the online setting is important in the decentralized FL scenario.
>
> **Response:** We believe online learning is important for federated learning scenarios. In many FL applications, optimization is a long-term process e.g., we need to update the model every day. In such cases, the distribution of data on each is varying with time. For example, users’ preference may also be affected by the current trend and thus is not static. Normal FL methods assume each client has a fixed distribution and are incapable of dealing with such cases. However, we can model them as online learning problems.

---

### Official Review · AnonReviewer3 · 2020-10-29
**This work focuses on single-sided decentralized federated learning. The authors propose a push sum-based algorithm to relax the symmetric matrix assumption, which leads to a flexible decentralized training on a directed network graph. The authors analyze the regret bound, as well as run some numerical experiments to demonstrate the correctness of the proposed algorithm.**

**Rating:** 8
**Confidence:** 5

**Review:**

This work focuses on single-sided decentralized federated learning. The authors propose a push
sum-based algorithm to relax the symmetric matrix assumption, which leads to a flexible
decentralized training on a directed network graph. The authors analyze the regret bound, as well as
run some numerical experiments to demonstrate the correctness of the proposed algorithm.
***Strengths***:
Overall the paper is well written.
1. The proposed method bridges a gap between existing decentralized federated learning algorithms
and real single-sided social networks. Another example I can recall is that sharing in the data market
is single-sided. The motivation is sound. ​As far as I know, this is the first paper in the FL community
talks about this setting.
2. The authors design a novel algorithm. Its theoretical results of the convergence rate ​connect
"online learning" and "asymmetric graphs", which is novel in federated learning.
3. The experimental design is excellent (including many settings). The code is very readable and
well-documented. I believe this helps the popularity of this proposed algorithm.
***Weakness***:
I can understand that this work focuses on the algorithm rather than provides a privacy guarantee.
So it would be great if the authors provide some intuitions or discussions about how to address
privacy concerns.
I prefer to demonstrate the proposed algorithm on more challenging datasets, but since this work
emphasizes the convergence analysis, it should be OK for me.
The authors only mention an important work in the related works section: “Notably, Zhao et al.
(2019) shares a similar problem definition and theoretical result as our paper. However, single-sided
communication is not allowed in their setting, restricting their results”. I suggest the authors discuss
more about it and distinguish the contributions in theory analysis.
Related works:
“Stochastic gradient push for distributed deep learning” (ICML 2019) should be discussed.
In section 4.4, more privacy-related works should be mentioned.
Overall Rating
Since the theory in this work is sound and the experimental design and code implementation are
also excellent, I incline to strongly support the acceptance of this work.

---

> ### Author Response · Authors · 2020-11-22
> **Response to Reviewer 3's comments**
>
> Thanks for your reviews of our paper. Below are our responses to your comment:
>
> > I can understand that this work focuses on the algorithm rather than provides a privacy guarantee. So it would be great if the authors provide some intuitions or discussions about how to address privacy concerns. I prefer to demonstrate the proposed algorithm on more challenging datasets, but since this work emphasizes the convergence analysis, it should be OK for me. The authors only mention an important work in the related works section: “Notably, Zhao et al. (2019) shares a similar problem definition and theoretical result as our paper. However, single-sided communication is not allowed in their setting, restricting their results”. I suggest the authors discuss more about it and distinguish the contributions in theory analysis. Related works: “Stochastic gradient push for distributed deep learning” (ICML 2019) should be discussed. In section 4.4, more privacy-related works should be mentioned. Overall Rating Since the theory in this work is sound and the experimental design and code implementation are also excellent, I incline to strongly support the acceptance of this work.
>
>
> **Response:** In fact, we have provided some intuitions about privacy protection in Section 4.4, though we don’t have a rigorous analysis of the privacy guarantee.
> As for the related works, we will add more discussion in the revision.

---

### Official Review · AnonReviewer4 · 2020-10-30
**Decentralized algorithm, but less consider the federated learning setting**

**Rating:** 4
**Confidence:** 4

**Review:**

## Summary

I do apologize for delaying the review process. I do spend lots of time and carefully read the paper. All comments listed below intend to help authors improve the quality of the manuscript. They are based on my understanding which might contain misunderstanding points if any. I hope comments are helpful and even the critiques are not discouraging your endeavor in the following.

First of all, the manuscript proposed an on-line push-sum algorithm to handle the decentralized SGD with the single-sided constrain. A rigorous regret analysis is provided for the proposed algorithms. The detailed comments are listed in the following.


## 1Major Comments:

- The motivation of the manuscript is really strange. The authors mentioned that they considered social network scenarios many times. However, the explanations and discussions are falling in the edge server setting.
- They proposed the online push-sum algorithm which is generalized from Tsianos (2012). I still cannot understand why the online push-sum algorithm is a federated learning algorithm. From my perspective, it is only a generalization of the push-sum algorithm in the online setting with single-sided trust constrain. Could you please show me the special feature of your setting?
- They provided a regret analysis of the proposed algorithm. However, the authors never showed that the contributions of the proof skill.
- In the simulation study, could you please show the network structure you proposed?

## 2 Minor Comments
- Page 2, Notation. You should introduce $n$ first, before the confusion matrix.
- Page 2, related work: the citation of Stich (2018) and Wang and Joshi (2018) is not corrected.
- Page 5, you should explain the “strongly connected” in Assumption 1 detailedly.

---

> ### Author Response · Authors · 2020-11-22
> **Response to Reviewer 4's comments**
>
> Thanks for your reviews of our paper. Below are our responses to your comments:
>
> ### Comment 1:
>
> >The motivation of the manuscript is really strange. The authors mentioned that they considered social network scenarios many times. However, the explanations and discussions are falling in the edge server setting.
>
> **Response:** Our proposed algorithm is very practical and has significant meaning in reality. We will highlight the significance more explicitly in our revision. Here are three concrete examples:
> 1) In social networks such as Facebook, although users are in the same group, they only share their personal information with friends who they trust;
> 2) In the financial system, especially blockchain-based decentralized ML, some information flow is also single-sided;
> 3) Taking the health data market Kara (https://kara.cloud/) as an example, users in Kara have ownership of their own private health records. Other users they shared with do not have an obligation to share data with them, or they may not have any intention to buy data from other users.
>
>
> ### Comment 2:
>
> >They proposed the online push-sum algorithm which is generalized from Tsianos (2012). I still cannot understand why the online push-sum algorithm is a federated learning algorithm. From my perspective, it is only a generalization of the push-sum algorithm in the online setting with single-sided trust constraints. Could you please show me the special feature of your setting?
>
> **Response:** Please refer to the definition of FL at “Advances and Open Problems in Federated Learning” (https://arxiv.org/pdf/1912.04977.pdf). This is a vision and review paper published by the original authors at Google who proposed the first FL algorithm FedAvg. In Section 2.1, this paper explicitly points out that fully decentralized/peer-to-peer distributed learning is an important scenario that FL targets. Our special features contain:
> 1. Our algorithm removes some constraints imposed by typical decentralized methods, making it more flexible in allowing arbitrary network topology. Each node only needs to know its out neighbors rather than the global topology;
> 2. We provide the rigorous regret analysis for the proposed algorithm and specifically distinguish two components in the online loss function: the adversary component and the stochastic component, which can model clients’ private data and internal connections between clients, respectively.
>
>
> ### Comment 3:
>
> >They provided a regret analysis of the proposed algorithm. However, the authors never showed the contributions of the proof skill.
>
> **Response:** Here we politely point out that the proof for our regret is more challenging because it allows each user in the network to hold different models, but still minimize the overall regret, even the network is single-sided. This is a more general case than the previous studies and we need to design a new proving strategy for the theoretical analysis.  The key idea is to prove that the gradient variance is reduced by using data from other users, when those data share some similarity. We will add more discussion about this part in our revision.
>
>
> ### Comment  4:
>
> > In the simulation study, could you please show the network structure you proposed?
>
> **Response:** The networks used in experiments are randomly generated. For each node, we first randomly generate an integer $b$ indicating the out-degree of this node. Then, we randomly sample $b$ nodes and add connections onto these nodes. In Section 5.4, we evaluate different network densities. Please check that sedition for details. If this is not adequate to convey our experimental design, we can provide the numerical matrix to represent our topology structure in our revision. Thanks for your suggestion for a better presentation.
>
>
>
>
> > Minor comments
>
> **Response:** Thanks for these comments, we will revise accordingly.

---

### Decision · Program_Chairs · 2021-01-08
**Final Decision**

**Decision:**

Reject

**Comment:**

The paper studies federated learning in what they call ```'single-sided trust' scenario, i.e. there is no dedicated server and the trust relationship is asymmetric.

This paper was a trickier case to decide on, and more borderline, in our opinion, than the reviewers' scores suggest, primarily, because the reviewers' recommendations are based on more subjective notions of novelty and importance/appropriateness of the studied setting, rather than identifying specific flaws in theoretical analysis or experiments. Ultimately, it boils down to three reviewers being (rather) negative about the paper, and the only (very) positive reviewer not stepping in to champion it. The negative reviewers believed that the novelty is not substantial enough to meet the high ICLR acceptance bar  (see details in reviews by R1 and R2 re similarity to ) and also have questioned the general motivation  (R1) and/or the online learning setting (R3).  While this assessment may be too harsh (esp. R1) - I think that the paper has merit - I share their feeling that in its current form it does not have a strong enough contribution.